# Enhancing trauma cardiopulmonary resuscitation simulation training with the use of virtual reality (Trauma SimVR): Protocol for a randomized controlled trial

Josef Michael Lintschinger[1,2], Philipp Metelka[1,2], Lorenz Kapral[1,2], Florian Kahlfuss[2], Lena Reischmann[1], Alexandra Kaider[3], Caroline Holaubek[2,4], Georg Kaiser[5], Michael Wagner[6], Florian Ettl[7], Leonhard Sixt[5], Eva Schaden[1,2], Christina Hafner[1,2,4]*

1 Ludwig Boltzmann Institute Digital Health and Patient Safety, Vienna, Austria, 2 Department of Anesthesia, Intensive Care Medicine and Pain Medicine, Medical University of Vienna, Vienna, Austria, 3 Center for Medical Data Science, Institute of Clinical Biometrics, Medical University of Vienna, Vienna, Austria, 4 Medical Simulation Center, Medical University of Vienna, Vienna, Austria, 5 Department of Orthopedics and Trauma Surgery, Medical University of Vienna, Vienna, Austria, 6 Department of Pediatrics, Comprehensive Center for Pediatrics, Medical University of Vienna, Vienna, Austria, 7 Department of Emergency Medicine, Medical University of Vienna, Vienna, Austria

* christina.hafner@meduniwien.ac.at

**Data Availability Statement:** No datasets were generated or analysed during the current study. All

# Abstract

## Background

With the increasing availability and use of digital tools such as virtual reality in medical education, there is a need to evaluate their impact on clinical performance and decision-making among healthcare professionals. The Trauma SimVR study is investigating the efficacy of virtual reality training in the context of traumatic in-hospital cardiac arrest.

## Methods and analysis

This study protocol (clinicaltrials.gov identifier: NCT06445764) for a single-center, prospective, randomized, controlled trial focuses on first-year residents in anesthesiology/intensive care, traumatology, and emergency medicine. The study will compare the clinical performance in a simulated scenario between participants who received virtual reality training and those who received traditional e-learning courses for preparation. The primary endpoint is the time to a predefined intervention to treat the underlying cause of the simulated traumatic cardiac arrest. Secondary endpoints include protocol deviations, cognitive load during simulated scenarios, and the influence of gender and personality characteristics on learning outcomes. The e-learning and the virtual reality training content will be developed in collaboration with experts from various medical specialties and nursing, focusing on procedural processes, guideline adherence specific to trauma patient care, and traumatic in-hospital cardiac arrest.

relevant data from this study will be made available upon study completion.

**Funding:** The author(s) received no specific funding for this work.

**Competing interests:** The authors have declared that no competing interests exist.

**Abbreviations:** ALS, Advanced life support; AOI, Area of interest; AWMF, Association of the Scientific Medical Societies in Germany; CPR, Cardiopulmonary resuscitation; CRF, Case report form; eFAST, Extended focused assessment with sonography for trauma; ERC, European Resuscitation Council; etCO2, End-tidal carbon dioxide; FiO2, Fraction of inspired oxygen; ID, Identification; IHCA, In-hospital cardiac arrest; MedUni Vienna, Medical University of Vienna; NASA TLX, National Aeronautics and Space Administration Task Load Index; ROSC, Return of spontaneous circulation; SBAR, Situation-Background-Assessment-Request; TEI, Training evaluation inventory; TCA, Traumatic cardiac arrest; VR, Virtual reality.

## Results

The results of this study will provide valuable insights into the efficacy of virtual reality training, contributing to the advancement of medical education, and serve as a foundation for future research in this rapidly evolving field.

## Introduction

Traumatic cardiac arrest (TCA) is a devastating condition with a low survival rate [1] that requires urgent recognition, rapid intervention, and specialized resuscitation skills in both in-hospital and out-of-hospital settings. To improve the survival from in-hospital cardiac arrest (IHCA), the International Liaison Committee of Resuscitation and the American Heart Association even recommend focused cardiopulmonary resuscitation (CPR) training as a primary area of interest [2, 3].

Simulation is the gold standard for training various medical scenarios in a safe and forgiving environment [4–6], with an additional focus on patient safety [5, 7]. This is particularly important in the context of rare but life-threatening conditions [4, 7, 8] such as TCA. In contrast, it is important to note that the highly specialized skills required to manage TCA may not be commonly taught in standard adult advanced life support (ALS) courses and simulations. In addition, the procedures required for TCA management deviate to some extent from standard ALS care, as suggested by the European Resuscitation Council (ERC) and are rarely used in daily practice. As a result, healthcare providers need to undergo rigorous training [9–11] to ensure that they have the necessary skills and expertise to effectively manage TCA patients when needed. Although simulation-based training is an effective method for improving trauma resuscitation skills, the complexity of TCA scenarios and the need for specialized equipment and trained personnel often restrict the availability of simulation opportunities, making them both costly and scarce. To address this issue, innovative, less expensive, and more accessible training methods need to be developed.

Virtual reality (VR) simulation is an emerging approach that enables realistic and immersive training scenarios without the need for expensive equipment or physical space [7, 12–14]. In-situ simulation, on the other hand, allows for training in the actual clinical environment, allowing for the assessment of team dynamics and human factors that may affect resuscitation outcomes. The combination of pre-course VR preparation [11, 15] and in-situ simulation approaches [11, 15–17] could improve the TCA management skills of health care providers and also have a positive impact on the outcomes of critically injured patients. Blended learning approaches appear to provide the best educational value for CPR training [11, 16, 18–20].

The primary aim of this study is to evaluate the efficacy of using VR technology, as opposed to e-learning [15, 21–23] preparation, to equip learners with the skills and knowledge necessary for in-person simulation training and to improve their performance in TCA management skills. The secondary aims are to assess gender differences in performance [24, 25], the frequency of protocol deviations, the cognitive load, and the gaze behavior between groups as well as participants' acceptance and impressions of VR in medical education.

We hypothesize that the use of VR in the context of trauma CPR training will lead to improved performance in the management of trauma patients going into cardiac arrest, with greater confidence in clinical decision-making, resulting in shorter times to order/perform predefined critical actions.

## Materials and methods

### Study design

This study is a single-center, prospective, single-blind, randomized, controlled trial focusing on first-year residents in anesthesiology/intensive care, emergency medicine, and traumatology at the Medical University of Vienna (MedUni Vienna).

This study protocol has been reported in accordance with the SPIRIT checklist (S1 File) and has been registered on clinicaltrials.gov with the following identifier number: NCT06445764.

Fig 1 shows details on the exact schedule of enrolment, interventions and the assessment while Fig 2 provides an overview of the study's timeline and events.

**Phase 1.** Recruited participants will watch a 60-minute video to learn about the requirements and steps of this study. Furthermore, this introductory video will give an overview of management strategies for trauma patients going into cardiac arrest in an in-hospital setting and provides links to the relevant guidelines [26–28] on which this study is based. Participants will then work independently through the guidelines provided. It also indicates that knowledge of the content of the guidelines is a prerequisite for phases 2 to 3. This video can be viewed as often as desired throughout the study. Participants will also be required to complete a questionnaire prior to randomization, which assesses participants' baseline characteristics (including the German version of the "openness to experience" subscale of the 100-item

| Activity/ Assessment | CRF (yes/ no) | Approximate time to complete | Study period | | | | | | | |
|---|---|---|---|---|---|---|---|---|---|---|
| | | | **Phase 1** | | **Phase 2** | | | | **Phase 3** | **Phase 4** |
| | | | $-t_0$ | $t_0$ | $t_1$ | $t_2$ | $t_3$ | $t_4$ | $t_5$ | $t_6$ |
| | | | Pre-study | Allocation | Training 1 | Training 2 | Training 3 | Training 4 | Assessment scenario | Video evaluation |
| Eligibility screen | No | n/a | X | | | | | | | |
| Informed consent | No | 20 min | X | | | | | | | |
| Introductory video | Yes | 60 min | X | | | | | | | |
| Questionnaire 1 | Yes | 10 min | X | | | | | | | |
| Randomization | Yes | n/a | | X | | | | | | |
| e-learning | Yes | 80 min | | | ◆——————◆ | | | | | |
| VR training | Yes | 100 min | | | X | X | X | X | | |
| Assessment | No | 20 min | | | | | | | X | X |
| Questionnaire 2 | Yes | 10 min | | | | | | | X | |

**Fig 1. Schedule of enrolment, interventions, and assessment.** $-t_0$ = recruitment process before the start of the study; $-t_0/t_0$ = phase 1; t1 –$t_4$ = 20-minute VR or e-learning training sessions within phase 2; $t_5$ = phase 3; $t_6$ = phase 4; *CRF = case report form, VR = virtual reality.*

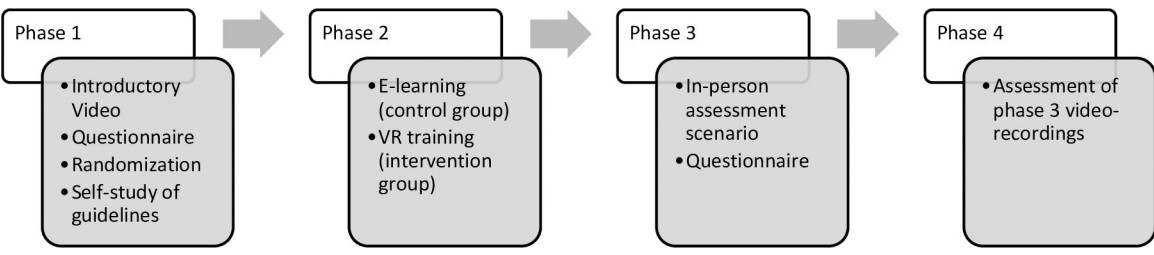

**Fig 2. Flowchart of the study.**

HEXACO-PI-R test [29, 30]). Subsequently, recruited participants will be randomly assigned to the control (e-learning course) or intervention group (VR training) on an equal basis by gender. Participants will not know their allocation to either group until they have watched the entire introductory video and completed the questionnaire. Randomization will be performed applying the minimization method [31] with gender as a stratifying factor. The web-based randomization software *Randomizer* from MedUni Vienna [32] will be used.

**Phase 2.** Participants will then complete either the e-learning course (control group) or the VR training (intervention group) over a two-week period, both focusing on the same content. Both groups will spend approximately the same amount of time learning the same content, which will prepare the participants for the assessment (phase 3). If participants do not complete the entire e-learning course or the entire VR training, they will be dropped from the study.

**Phase 3.** Within one week of completing the VR- or e-learning preparation courses, participants will attend an in-person assessment session and will be evaluated in the role of medical team leader of a standardized TCA scenario. These sessions will take place at the Medical Simulation Center of MedUni Vienna.

In addition, each participant will be required to complete another questionnaire immediately following the assessment session (including the German version of the training evaluation inventory (TEI) [33, 34]).

**Phase 4.** Phase 3 assessment sessions will be recorded on video and evaluated by unbiased and blinded experts.

## Objectives and outcomes

Table 1 provides an overview of the primary and secondary objectives of this study, together with the corresponding outcome measures.

## Recruitment and eligibility criteria

**Inclusion criteria.** In order to be considered eligible for this study, participants must satisfy all of the following inclusion criteria:

- First-year resident in the field of anesthesiology, intensive care, emergency medicine, or traumatology at MedUni Vienna

- ≥ 18 years

**Exclusion criteria.** Participants who meet any of the following criteria will not be included in the study:

- Pre-disposition for cybersickness (motion sickness, pregnancy, pre-existing cybersickness)

**Table 1. Study objectives.**

| Objective | Objective definition | Measure | Measure definition |
|---|---|---|---|
| Primary | To evaluate the difference in time to critical action #2 between randomized groups | Time (seconds) | Expert assessment of video recordings of the assessment session: time from the onset of cardiac arrest to critical action #2 (as defined in Table 2) |
| Secondary | To evaluate the difference in time to critical action #1 between randomized groups | Time (seconds) | Expert assessment of video recordings of the assessment session: ▪ Critical action #1: time from the scenario start to the recognition of the underlying cause (as defined in Table 2) |
| Secondary | To evaluate the difference in unrecognized/ untreated underlying causes of TCA between randomized groups | Critical action performed (yes/no) | Expert assessment of video recordings of the assessment session: one or more missed critical actions (as defined in Table 2) |
| Secondary | To evaluate the difference in the number of patients declared dead prematurely between the randomized groups | Patient declared to be dead before the 15-minute time limit of the scenario (yes/no) | Expert assessment of video recordings of the assessment session |
| Secondary | To evaluate the difference in frequency of protocol deviations between randomized groups | Protocol deviation (yes/no) | Expert assessment of video recordings of the assessment session: protocol deviations (as defined in Table 3) |
| Secondary | To evaluate gender differences in the primary outcome | Time (seconds) | Expert assessment of video recordings of the assessment session: time from the onset of cardiac arrest to critical action #2 (as defined in Table 2), based on gender |
| Secondary | To evaluate the difference in cognitive load between randomized groups | 0 to 100 points | Assessed immediately following the assessment session ▪ Global NASA TLX score ▪ NASA TLX score per objective (mental demand, physical demand, temporal demand, performance, effort, frustration) |
| Secondary | To evaluate the difference in gaze behavior using eye-tracking technology between randomized groups | ▪ dwell-time in AOIs (seconds) ▪ fixation count in AOIs ▪ average fixation duration in AOIs (seconds) ▪ time when no AOI is illustrated (seconds) | Tobii Pro glasses analyzer software; assessment based on recordings of the assessment sessions |
| Secondary | To evaluate the incidence rate of VR-related adverse events | Adverse events (yes/no) | Incidence of nausea, vomiting, dizziness, headache, overexertion/ fatigue of the eyes (discomfort, blurred vision), stumbling, falling, bumping into real world objects |
| Secondary | To evaluate the difference in learning outcomes and training design using the TEI between randomized groups | 5-point Likert scales | Training outcomes (subjective fun, perceived usefulness, perceived difficulty, subjective knowledge growth, attitude towards training) and training design (problem-based learning, activation (of prior knowledge), demonstration, application, integration) will be measured using a 45-items inventory (I do not agree; I rather not agree; neither nor; I rather agree; I agree very much) |

AOI = area of interest, CPR = cardiopulmonary resuscitation, NASA TLX = National Aeronautics and Space Administration Task Load Index, TEI = training evaluation inventory, TCA = traumatic cardiac arrest, VR = virtual reality.

## Sample size

The time to critical action #2 is considered as primary outcome measure. Time values of participants who do not identify and treat the underlying condition, or who declare dead prematurely, will be set to 15 minutes and considered as censored observations. Therefore, a non-parametric test was used for sample size calculation, and the probability that an observation in one group will be less than an observation in the other group is considered as the effect size measure. With a sample size of 30 in each group the two-tailed Wilcoxon (Mann-Whitney) rank-sum test will have 80% power to detect a probability of at least 0.71, that an observation in one group is less with respect to the primary outcome variable than an observation in the other group. A two-sided significance level of 0.05 is considered. The software nQuery (nQuery 9, ©Statistical Solutions Ltd. 2024) was used for sample size calculation.

Based on an expected drop-out rate of 10%, 67 participants will be recruited to participate in this study.

## Control and intervention

Both learning methods have a shared limitation in that they cannot provide hands-on practice for manual skills. Consequently, the content of these courses will primarily concentrate on procedural processes and adherence to guidelines within the realm of trauma patient care and traumatic IHCA.

Participants will engage in case-based learning using a variety of case vignettes offered in both e-learning and VR environments. The cases used will focus on the same medical content and are designed to be practiced in a structured way. The overarching goal of both learning modalities is to *"apply"* the necessary life-saving action sequences within the framework of a TCA, according to Bloom's taxonomy [33, 34].

The key features of each learning modality are outlined below:

**E-learning**: This course includes step-by-step descriptions of action sequences, along with videos demonstrating specific actions and communication examples. Interactive components–such as mapping exercises, quizzes, and comprehension questions–help reinforce learning. Participants can explore images with expandable content, follow action sequences in chronological order, and respond to questions that encourage both clinical decision-making and self-reflection.

**VR**: In this interactive virtual environment, participants play through scenarios, assign specific tasks to virtual team members, and make clinical decisions. Virtual team members provide communication examples with feedback. At the end of each scenario, participants receive feedback on their adherence to guidelines, along with questions for self-reflection.

The VR environment as well as the e-learning course is being developed by the research team and industry partners, with additional expertise from specialists in anesthesia, intensive care, emergency medicine, traumatology and nursing. To ensure optimal efficiency during the VR training, an additional mandatory orientation module of 20 to 30 minutes, to be played once before using the VR scenarios, will be provided to all participants in the VR group. This module aims to familiarize participants with the technical features of the VR headset and controllers. By doing so, participants can maintain their focus on the medical content rather than being distracted by technical aspects. A head-mounted display (HTC Vive Focus 3) will be used for the single-player VR scenarios, with an additional trainer operating from a computer that serves as the control unit to manage the scenarios.

**VR-related adverse events.**   During the utilization of VR, adverse events may potentially arise, with cybersickness [35] being a particularly common concern. The study will assess the occurrence of specific symptoms, including nausea and/or vomiting, dizziness, headaches, eye strain, and fatigue (such as discomfort, blurred vision, or reduced visual acuity), as well as physical effects such as tripping, falling, or colliding with real-world objects. The incidence of these symptoms will be monitored and evaluated as part of the study's analysis.

## Assessment sessions

Prior to commencing the manikin-based simulation scenario, participants will be given 15 to 20 minutes to familiarize themselves with the room, the environment, the manikin's functions, equipment, and provided medications. In the assessment scenario, all participants will use the same male adult manikin, which will display clinical signs such as cyanosis, breath sounds, and pulses to aid in clinical decision-making. An instructor will be present to assist the group during this phase.

Both groups will face the same standardized assessment scenario: a trauma patient with a single predefined underlying condition that progresses to an IHCA after a predefined short latency period. Each scenario will have a duration of 15 minutes.

To ensure an objective evaluation of participants' performance, the scenarios will be recorded using cameras that are preinstalled in the simulation room. Furthermore, all actions will be captured from the participant's perspective using Tobii Pro 3.0 eye-tracking glasses. The simulation room is designed to replicate the conditions and materials found in a real emergency room, providing high fidelity. Expert members of the study team will control the high-fidelity manikin in a predetermined and standardized manner from the control room.

During the assessment, the participant being evaluated will enter the scenario individually. The scenario involves a team of seven individuals, with the team leader being the sole focus of evaluation. The remaining six team members, confederates of the study team, take on the roles of a standardized emergency team but are strictly instructed not to assist the team leader in making medical decisions. They can only perform actions upon receiving orders from the team leader.

Before the scenario commences, all relevant information regarding the patient's rescue by the emergency medical service will be provided to the team leader (participant) on a sheet of paper. This information is presented using the SBAR (Situation-Background-Assessment-Recommendation) approach, ensuring a concise and structured communication framework. The scenario begins precisely at the moment of the "start" signal, exactly 20 seconds after the participant has received the aforementioned sheet of paper.

## Performance assessment

To evaluate participants' adherence to established guidelines and protocols in the assessment scenario, critical actions (outlined in Table 2) and key actions required for managing a TCA, along with potential deviations (outlined in Table 3), were defined. These criteria are based on the guidelines provided by the ERC for adult ALS [26], cardiac arrest in special circumstances [27], and the AWMF (Association of the Scientific Medical Societies in Germany) guideline on polytrauma and management of critically injured patients [28].

Critical actions are defined as actions that, if not performed correctly and in a timely manner, could potentially cause significant harm to the patient in the scenario. These critical actions are interconnected, and the primary outcome is, therefore, the time taken until initiating the most critical action in the assessment scenario (critical action #2). As the assessment scenario is based on a single underlying medical condition, other possible simultaneous necessary actions are not addressed in the context of performance assessment.

The assessment scenario will end at exactly 15 minutes with a return of spontaneous circulation (ROSC) achieved if all critical actions have been completed before the allotted time has elapsed.

**Table 2. Critical actions.**

| | Critical action | Time measurement | Notes |
|---|---|---|---|
| #1 | Recognition of the underlying cause | From the scenario start to recognition of the underlying cause | The defined point in time measurement is when the team leader fully pronounces words or phrases from which the assessor can conclude the diagnosis. These words or phrases will be documented. |
| #2 | Predefined therapeutic intervention (e.g. mini-thoracotomy, pelvic splint) | From the onset of cardiac arrest to performing the predefined therapeutic intervention | The time at which a team member's hands or medical instrument first touch the patient to perform this predefined therapeutic intervention is the defined point for time measurement |

CPR = cardiopulmonary resuscitation.

**Table 3. Protocol deviations.**

| Category | | Actions/orders |
|---|---|---|
| **Primary assessment (xABCDE approach)** | **x** | ■ Actively searching for signs of massive internal or external bleeding |
| | **A/ B** | ■ Check airway/tube placement<br>■ Check thorax<br>■ Check respirator settings ■ Check jugular veins<br>■ Check $spO_2$/$etCO_2$ |
| | **C** | ■ Pulse check/heart rate<br>■ Check of skin color<br>■ Check of capillary refill time<br>■ Check of blood pressure<br>■ At least one big lumen vascular access |
| | **D** | ■ Pupil check<br>■ Re-Evaluation/anticipation of analgesic medication<br>■ Re-Evaluation/anticipation of narcotic medication |
| | **E** | ■ Removing clothes<br>■ Temperature check |
| **Secondary assessment** | | ■ Full body examination<br>■ Point-of-care ultrasound examination<br>■ Computer tomography scan |
| **Advanced life support** | | ■ CPR start (initiation within 10 seconds following the onset of cardiac arrest, or within 10 seconds after performing prioritized measures [e.g. mini-thoracotomy])<br>■ Time keeping<br>■ Chest compressions depth (feedback)<br>■ Chest compressions rate (feedback)<br>■ Exchange of person performing chest compressions every two minutes<br>■ Defibrillator placement<br>■ Rhythm analysis at correct times<br>■ Correct interpretation of all rhythms in analysis phases<br>■ Pulse checks at correct times<br>■ Continuous ventilation while chest compressions in correct rate (10/min) or 30:2 algorithm<br>■ $FiO_2$ set to 1.0<br>■ Medications<br>■ Arterial blood gas (if not already done in the primary/secondary assessment)<br>■ Check of possible reversible causes |

CPR = cardiopulmonary resuscitation, $etCO_2$ = end-tidal carbon dioxide, $FiO_2$ = fraction of inspired oxygen, ROSC = return of spontaneous circulation.

If participants fail to identify the underlying cause of the simulated TCA and subsequently fail to prioritize and perform the predefined therapeutic intervention for the scenario, a simulated specialist will enter the scenario and take over team leadership at the exact 15-minute mark. In such cases, the scenario will proceed for an additional three minutes to achieve a ROSC and still ensure a positive learning outcome for the participants. If a patient is declared dead prematurely (before the 15-minute time limit has elapsed), the scenario will be terminated, and a standardized full-team debriefing session will be conducted. However, debriefing will be carried out after each assessment scenario, irrespective of the patient's outcome.

Table 3 provides a comprehensive compilation of items considered protocol deviations until the required action is explicitly carried out or verbally communicated in its entirety. To be deemed timely within this context, action or order must be accomplished within the corresponding chronological category outlined in Table 3. Each protocol deviation can occur only once, and actions ordered or performed by the team leader are considered equivalent in this

context. It is worth noting that, within the scope of this study, all orders and actions can be executed successfully on the first attempt, contributing to the standardization of the scenario.

The video recordings will be subjected to analysis by two blinded experts in anesthesiology and intensive care, who also hold an additional diploma in emergency medicine. These experts are based at the Department of Anaesthesia, Intensive Care Medicine and Pain Medicine at MedUni Vienna. They will not be involved in the implementation of the study and will remain unaware of the group assignments from phase 2. To minimize inter-rater variability, experts will independently assess four exercise cases specifically selected to include two cases designed to be easy and two cases designed to be challenging in advance. The results will be subsequently discussed among the experts and the research team, allowing for the resolution of any ambiguities or disagreements and reaching a consensus for further evaluation.

**Cognitive load.**  Immediately following the assessment scenarios, the cognitive load experienced by participants will be assessed using the National Aeronautics and Space Administration Task Load Index (NASA TLX) questionnaire [36], which is a widely recognized and established gold standard tool for measuring subjective workload. The questionnaire incorporates ratings on six sub-dimensions: mental demand, physical demand, temporal demand, performance, effort, and frustration. Participants will provide ratings on each sub-dimension, which range from 0 to 100. Additionally, the global Task Load Index, which represents the mean value of all sub-dimensions, will be calculated.

**Eye-tracking.**  In phase 3, differences in gaze behavior between the randomized groups will be examined by using eye-tracking technology. The parameters assessed include the dwell time on different areas of interest (AOIs), the number of fixations, and the average duration of fixations. Specifically, the defined AOIs for analysis are the manikin's head/airway, the manikin's thorax, the vital signs monitor, the ventilator monitor, *and team members hands or body*.

The number of fixations refers to the instances when participants shift their gaze between different AOIs, while the average fixation duration represents the average time spent fixating on a specific AOI. A longer average fixation duration indicates a higher level of cognitive engagement with that AOI.

Additionally, the analysis will consider the relative time when no AOI is in focus, providing insights into periods of visual disengagement or lack of specific focus during the scenario. These gaze behavior metrics will comprehensively evaluate participants' visual attention and cognitive processing during the assessment. Two members of the study team, who are experienced in eye-tracking analyses, will conduct the eye-tracking analyses. Moreover, prior to the commencement of each assessment scenario, the eye-tracking glasses will be calibrated on the respective participant. Additionally, attention will be paid to ensure that the lighting and environmental conditions are as identical as possible, thus ensuring optimal eye-tracking quality in terms of spatial precision and spatial accuracy.

## Data management and confidentiality

Data will be collected from the expert-based video assessments, from the Tobii Pro glasses analyzer software and questionnaires from phases 1 and 3, including variables to capture participants' baseline characteristics. These variables include age (in years), gender (male/female/diverse/inter/open/none), German language skills (first language/fluent/basic), clinical expertise (first-year resident in anesthesia and intensive care medicine/traumatology/emergency medicine), previous participation in ERC ALS provider courses (yes/no), previous participation in certified advanced trauma courses (such as pre-hospital trauma life support, European trauma course, advanced trauma life support; yes/no), real-world experience with in-hospital CPR (rated on a 5-point Likert scale), real-world experience with out-of-hospital CPR (rated

on a 5-point Likert scale), real-world experience with in-hospital trauma care (rated on a 5-point Likert scale), real-world experience with out-of-hospital trauma care (rated on a 5-point Likert scale), openness to experience (HEXACO PI-R subscale; rated from 1 to 5), as well as participants' self-reported VR- and non-VR video game skill sets (rated on 5-point Likert scales).

The data collected will be securely stored at the study site. Any modifications made to the database will be carefully documented to ensure transparency and accountability. Data will only be shared with the Institute of Clinical Biometrics at MedUni Vienna in pseudonymized form for statistical evaluation.

## Statistical methods

Statistical analysis will be conducted using IBM SPSS (© IBM Corp. Released 2023. IBM SPSS Statistics for Macintosh, Version 29.0 Armonk, NY: IBM Corp) and SAS version 9.4 (© SAS Institute Inc. 2020, Cary, NC, USA).

**Descriptive analyses.** Counts and percentages will be given for categorical data. Continuous data will be reported using median and interquartile range if skewed or mean ± standard deviation if normally distributed. Boxplots will be used to visualize the distributions of continuous variables.

**Primary objective/outcome.** The primary outcome time to critical action #2 will be compared between the two randomized groups. Participants who do not identify and treat the underlying condition of TCA will be censored at 15 minutes. Furthermore, if patients are declared dead prematurely, the observation will also be censored at 15 minutes. To allow for censoring in this outcome variable the two-sided generalized Wilcoxon test will be used for comparison. The median times (with 95% confidence intervals) will be estimated using the Kaplan-Meier method. Univariate and multivariable Cox regression models will be used to evaluate the group effect (intervention vs control) and the gender effect on the primary outcome. The potential modifying effect of gender on the group effect will be tested by inclusion of an interaction term. The HEXACO subscale will also be included in the multivariable model to adjust for potential imbalances between the randomized groups. Hazard ratios (with 95% confidence intervals) will be presented to quantify the strength of these effects.

**Secondary objectives/outcomes.** A two-sided generalized Wilcoxon test will be used to compare times to critical actions (excluding the primary outcome), between the two randomized groups. Again, if participants do not identify and treat the underlying condition of TCA, or if patients are declared dead prematurely, the times will be censored at 15 minutes. Median times (with 95% confidence intervals) will be calculated by the Kaplan-Meier method. We will compare the number of participants who did not recognize and treat the underlying condition of the TCA, using the two-sided chi-square test. The same test will be conducted to determine the number of participants who have declared the patient dead prematurely. Odds ratios (with 95% confidence interval) will be calculated to quantify the strength of these group differences.

Each protocol deviation listed in Table 3 will be analyzed separately. The respective frequencies will be compared between groups calculating the two-sided chi-square test. Two-sided Fisher's exact test will be used if one or more expected cell frequencies are less than 5.

The two-sided Wilcoxon rank sum test will be used to compare the gaze behavior (dwell-time in AOIs, fixation count and average fixation duration, and relative time when no AOI is illustrated) and the NASA TLX scores (global and per objective) between randomized groups. In case of normal distributions in these secondary outcome measures the two-sided independent samples t-test will be used. 5-point Likert scales will be reported using descriptive statistics only.

Observations with missing data in the primary outcome measure will be excluded from the statistical analyses. For secondary outcome measures, the number and reasons for missing values will be reported. Although the prospective design of the study minimizes the likelihood of missing data in secondary outcomes, analyses of these measures will be conducted only if data is available for more than 80% of participants.

Two-sided p-values of < 0.05 will be considered statistically significant. Due to the singular primary objective, no correction for multiple testing will be made. All secondary objectives are exploratory in nature and will be used to generate further hypotheses.

### Ethics and safety considerations

The study will be conducted in accordance with the latest revision of the *Declaration of Helsinki for Biomedical Research Involving Human Subjects* and the current *International Conference on Harmonization—Good Scientific Practice guidelines* of MedUni Vienna.

The current protocol (S2 File), the associated questionnaires, and the informed consent form have been approved by the ethics committee of MedUni Vienna under the identification number 1388/2023. This study will also be reviewed by the data protection commission of MedUni Vienna.

Before being included in the study, potential participants will receive detailed information regarding the study's purpose, procedures, data used, and potential risks involved. Furthermore, as this is only a comparative study of different learning/training methods, there is hardly direct risk to participants. One risk is the disclosure of sensitive data. However, this will be mitigated by pseudonymizing and anonymizing the data collected and restricting access to it. Recorded videos will be deleted after final evaluation in accordance with data protection regulations. Another risk associated with this study is the occurrence of cybersickness, which can occur very rarely when using VR. This would usually improve shortly after removing the virtual reality headset.

Participants will be informed that their participation in the study is voluntary and that they have the right to withdraw from the study at any time and without giving any reason.

If participants choose to participate, they will be required to provide their informed consent by signing a consent form.

### Discussion

The Trauma SimVR study has been designed to meet the demand for innovative digital training tools in medical education. It utilizes VR training alongside an objective assessment tool to evaluate its efficacy specifically within the context of IHCA. This area of research is of high importance since there is a growing presence of digital tools, such as VR, augmented/mixed reality, serious video games, chatbots, and others, being utilized in both clinical settings and medical education. However, the effectiveness of these tools in enhancing healthcare professionals' performance remains largely unexplored. Therefore, the present study aims to take a step forward by focusing specifically on VR-supported medical education for rare and complex trauma emergency scenarios.

In addition, it is important to highlight that there is currently a lack of validated scores or similar measures for assessing performance in scenarios like the ones addressed in this study. Consequently, the chosen approach of "time to critical action", which has been previously utilized [37, 38], was selected as the primary outcome measure. It is also essential to recognize that neither VR nor e-learning can fully replicate comprehensive clinical training. Therefore, the exploratory evaluation of protocol deviations could serve as a foundation for future studies, determining which areas may be effectively trained using digital approaches.

Furthermore, special attention will be given to examining the influence of gender and personality structure on learning outcomes. By understanding how different factors such as gender and personality traits influence learning outcomes, tailored approaches could be designed to maximize the effectiveness of educational interventions.

## Study limitations

This study aims to explore how effective VR training is in improving performance in managing traumatic IHCA care. However, it is still unclear how well the skills and competencies gained from such training can be applied to real-life situations. More research is needed to examine outcomes, such as mortality rates, for patients treated by healthcare professionals who have been trained using VR, in order to address this gap.

Due to the complexity of managing TCAs and the demands of leadership in critical situations, the impact of a single training period on skill retention remains unclear. Further research is needed to determine the optimal duration and frequency of VR training sessions to support long-term skill retention effectively.

Moreover, the scenarios in this study are specific to trauma patients in an emergency department setting. While this approach offers advantages, allowing for the practice of rare and resource-intensive scenarios in simulation training, the generalizability of the findings to other healthcare contexts needs to be further investigated.

It is important to acknowledge that the acquisition of VR equipment and the development and procurement of VR software may pose financial challenges for less economically privileged healthcare facilities. However, this also presents an opportunity for more resource-efficient training compared to expensive simulation centers. Further studies are required to evaluate the precise cost savings associated with this approach.

Finally, it is worth mentioning that digital learning currently has limitations, such as the practice of communication strategies or manual procedures including haptic feedback. Therefore, blended learning approaches [11, 15–20], of which VR can be a part, may offer the greatest benefits. Further research into these approaches will contribute to a more comprehensive understanding of their benefits and potential drawbacks.

## Supporting information

**S1 File. SPIRIT checklist.**
(PDF)

**S2 File. Study protocol approved by the ethics committee.**
(PDF)

## Author Contributions

**Conceptualization:** Josef Michael Lintschinger, Philipp Metelka, Lorenz Kapral, Florian Kahlfuss, Lena Reischmann, Caroline Holaubek, Georg Kaiser, Michael Wagner, Florian Ettl, Leonhard Sixt, Christina Hafner.

**Methodology:** Josef Michael Lintschinger, Alexandra Kaider, Christina Hafner.

**Project administration:** Josef Michael Lintschinger.

**Supervision:** Eva Schaden, Christina Hafner.

**Writing – original draft:** Josef Michael Lintschinger.

**Writing – review & editing:** Josef Michael Lintschinger, Philipp Metelka, Lorenz Kapral, Florian Kahlfuss, Lena Reischmann, Alexandra Kaider, Caroline Holaubek, Georg Kaiser, Michael Wagner, Florian Ettl, Leonhard Sixt, Eva Schaden, Christina Hafner.

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
