## [Decision Letter · Decision Letter 0]

5 Sep 2024

PONE-D-24-24609Enhancing trauma cardiopulmonary resuscitation simulation training with the use of virtual reality (Trauma SimVR): Protocol for a randomized controlled trial

PLOS ONE

Dear Dr. Hafner,

Thank you for submitting your manuscript to PLOS ONE. After careful consideration, we feel that it has merit but does not fully meet PLOS ONE’s publication criteria as it currently stands. Therefore, we invite you to submit a revised version of the manuscript that addresses the points raised during the review process.

We look forward to receiving your revised manuscript.

Kind regards,

Anne Lee Solevåg, M.D., Ph.D.

Academic Editor

PLOS ONE

Journal requirements: 1. When submitting your revision, we need you to address these additional requirements. Please ensure that your manuscript meets PLOS ONE's style requirements, including those for file naming. The PLOS ONE style templates can be found at https://journals.plos.org/plosone/s/file?id=wjVg/PLOSOne_formatting_sample_main_body.pdf and https://journals.plos.org/plosone/s/file?id=ba62/PLOSOne_formatting_sample_title_authors_affiliations.pdf. 2. When completing the data availability statement of the submission form, you indicated that you will make your data available on acceptance. We strongly recommend all authors decide on a data sharing plan before acceptance, as the process can be lengthy and hold up publication timelines. Please note that, though access restrictions are acceptable now, your entire data will need to be made freely accessible if your manuscript is accepted for publication. This policy applies to all data except where public deposition would breach compliance with the protocol approved by your research ethics board. If you are unable to adhere to our open data policy, please kindly revise your statement to explain your reasoning and we will seek the editor's input on an exemption. Please be assured that, once you have provided your new statement, the assessment of your exemption will not hold up the peer review process. 3. Your ethics statement should only appear in the Methods section of your manuscript. If your ethics statement is written in any section besides the Methods, please move it to the Methods section and delete it from any other section. Please ensure that your ethics statement is included in your manuscript, as the ethics statement entered into the online submission form will not be published alongside your manuscript.  4. Please include captions for your Supporting Information files at the end of your manuscript, and update any in-text citations to match accordingly. Please see our Supporting Information guidelines for more information: http://journals.plos.org/plosone/s/supporting-information. 

Reviewers' comments:

Reviewer's Responses to Questions

**Comments to the Author**

1. Does the manuscript provide a valid rationale for the proposed study, with clearly identified and justified research questions?

Reviewer #1: Yes

Reviewer #2: Partly

2. Is the protocol technically sound and planned in a manner that will lead to a meaningful outcome and allow testing the stated hypotheses?

Reviewer #1: Yes

Reviewer #2: Partly

3. Is the methodology feasible and described in sufficient detail to allow the work to be replicable?

Reviewer #1: No

Reviewer #2: No

4. Have the authors described where all data underlying the findings will be made available when the study is complete?

Reviewer #1: Yes

Reviewer #2: Yes

5. Is the manuscript presented in an intelligible fashion and written in standard English?

Reviewer #1: Yes

Reviewer #2: No

6. Review Comments to the Author

You may also provide optional suggestions and comments to authors that they might find helpful in planning their study.

Reviewer #1: Simulation training is regarded as a major element of patient safety and preparation for time-critical life-threatening medical emergencies, i.e. severe trauma and cardiac arrest. Despite the high benefit of in-situ simulation training in such scenarios, it is rarely carried out due to the complexity of the scenarios to be practiced and the high equipment and personnel expenditure required.

Extended reality (XR) is becoming increasingly important in medical simulation training as it allows immersive realistic training scenarios undertaken with less expensive equipment. Virtual reality (VR) is currently the most common variant of XR in this context.

The presented study protocol intends to compare the effectiveness of VR-based and E-learning-based preparation for in-situ training of in-hospital traumatic cardiac arrest (TCA)- a rare, but extremely challenging, scenario. The hypothesis is that this blended-learning approach is of superior educational value and could improve the management skills in TCA.

Strengths of the paper:

The criteria for recruitment and eligibility, the randomization procedure, the assignment of the participants to the comparison groups, the primary and secondary endpoints of the study and the chosen measurement methods are well chosen und clearly described. The secondary endpoints “influence of gender and personality” could give valuable insight into potential additional requirements or needs of important user subgroups. Tables and figures are clear and easy to interpret. The SPIRIT checklist was used to report all study protocol items. The protocol fulfills the respective legal, ethical and security requirements, was approved by the local ethics committee and the study has been registered in clinicaltrial.gov. In addition the protocol will also be reviewed by the data protection commission. The statistic procedures are well described, suitable and possible confounders are adequately taken into account. The cited literature sources are appropriate and mostly up to date.

Points where I would like to suggest a revision:

The authors state that (Page 3, lines 81 ff) the preparation training will “equip learners with the skills and knowledge necessary for in-person simulation training and to improve their performance in TCA management skills”. Unfortunately, only scarce information is provided about the specific learning objectives and contents, as well as the media technology and didactic implementation of the e-learning and VR training. The presentation of these points is particularly important because the treatment of traumatic cardiac arrest is a very complex task that requires a) theoretical knowledge of the mechanisms and treatment principles of trauma and resuscitation, b) manual or technical skills to eliminate the underlying causes, e.g. tension pneumothorax, and c) excellent management or non-technical skills. This makes it difficult to answer the questions whether learning objectives and content of the two methods are comparable. I therefore would recommend to describe the intended interventions in more detail. More details would also enhance the understanding of the setting of the in-situ simulation training (details see below).

Related to the chosen endpoints and their significance I would suggest a more detailed discussion of potential limitations (details see below).

Page 5, lines 115 ff:

“Participants will then complete either the e-learning course (control group) or the VR training

(intervention group) over a two-week period, both focusing on the same content”.

E-Learning:

Which kind of media are used? As trauma shockroom management depends not only on clinical skills but also on communication and management skills: To what extent are these “non-technical skills” addressed? Does the e-learning contain intervening questions that encourage participants to self-reflect? Are there other interactive elements, i.e. clicking on answer options, e.g. to select a cause of cardiac arrest or to decide on a specific diagnostic or therapeutic measure, or abstracted manual actions such as applying a pelvic sling, insertion of a chest tube?

VR training:

Will a head-mounted display (HMD) be used? If so, what type? Can you present the technical specs (i.e. as supplement data)? Does the VR “training” represent training in the narrower sense of the word, i.e. is it interactive and allows active clinical decisions, or is does it consist of mere passive tracking of the content presented?

If the former is true, could you describe the types of interaction?

Page 6, lines 122 ff:

How long is the interval between completion of Phase 2 and Start of Phase 3?

Page 7, Table 1:

Several secondary endpoints are measured to assess the participant´s subjective impressions and attitudes (i.e. level of enjoyment, learning process, self-confidence). Are these measurements based on validated test instruments? If not, what reasons spoke against? There are widely recognized assessment tools in this area, i.e. TEI (Training Evaluation Inventory).

Secondary objective: “Critical action #1: time from the onset of cardiac arrest to the start of chest

compressions (as defined in Table 2)”.

According to current resuscitation guidelines, chest compression is of secondary importance to other measures in traumatic cardiac arrest: German S3 updated guideline on the treatment of severe/multiple injuries: “Recommendation 2.3.4 [translated into English]: “Resuscitation of traumatic cardiac arrest should focus on the immediate, simultaneous treatment of reversible causes and takes priority over chest compressions”. ERC guidelines 2021 (Lott et al.): „In cardiac arrest caused by hypovolaemia, cardiac tamponade or tension pneumothorax, chest compressions are unlikely to be as effective as in normovolaemic cardiac arrest and may reduce residual spontaneous cardiac output. Therefore, chest compressions take a lower priority than addressing the reversible causes. Chest compressions must not delay immediate treatment of reversible causes.”

Could this potentially limit the significance of the primary endpoint “start of chest compressions”?

Page 7, Table 1: Secondary objectives: “To evaluate the difference in gaze behavior using eye-tracking technology between randomized groups”.

How is the quality of eye tracking (i.e. spatial precision and spacial accuracy) ensured?

Page 8, line 151: Inclusion criteria: “Only people who do not need eyeglasses for using VR”

How can this be interpreted and how do you practically handle this problem? Do interns wearing eye glasses and willing to participate in the study first test a VR system / HMD before they definitively decide to participate or to withdraw from the trial? Since the high percentage of people dependent on visual aids, this is not a mere theoretical question….

Page 9, lines 170-171: Additional VR orientation module: What is the duration of this module? Will it be mandatory for all participants In the VR group?

Page 9, lines 184-186:

Is the duration of the familiarization phase standardized? Do the participants familiarize themselves individually or as a whole group? Is there an instructor who assists during this phase i.e. explaining the use of the manikin functions?

Page 9, lines 187-188 and page 10, lines 200-205:

Does the virtual patient in the scenario already present with a cardiac arrest (i.e. at the moment of the virtual hand-over by the EMS team) or does the cardiac arrest develop after a predetermined short latency?

Medical scenarios: Is there always a single injury/medical problem leading to the cardiac arrest or maybe also a combination of underlying causes?

Does the “high-fidelity” manikin used in the study display clinical signs like cyanosis, recap time, breath sounds, subcutaneous emphysema, obstructed airway (which could facilitate clinical decision making)?

Is a single type of manikin used or do the manikins vary, i.e. according to scenario? What age group (Child? adolescent? adult? Young adult? old adult?) and sex does the manikin used in the study represent?

Page 10, lines 215-217:

“These critical actions are interconnected, and the primary outcome is, therefore, the time taken to execute the most critical action in the assessment scenario (critical action #3).”

In practice, when carrying out life-saving measures, it will take some time until a) they have been carried out after being initiated and b) until the success of the measures can be assessed. Only then can further decisions be made and further measures ordered. Consecutively

a) the term “time taken to execute” may be difficult to interprete or be misleading and

b) the problem of eventually simultaneously necessary actions is not addressed nor solved

page 13: Line 264-266:

“Specifically, the defined AOIs for analysis are the manikin's head/airway, the manikin's thorax, the vital signs monitor, and the ventilator monitor”.

The AOI mentioned are certainly important. However, the role of the trauma leader also includes distributing tasks sensibly within the team and supervising the respective team members. Therefore, it may also be useful to take into account the frequency and duration of the traumaleader´s gaze on team members, i.e. in order to recognize inappropriate actions or “underemployment”.

Page 17, lines 370 ff: Study limitations:

Traumatic cardiac arrest is an extremely demanding emergency, the successful management of which requires a high level of clinical expertise and the leadership and cooperation of a large interdisciplinary and interprofessional team. The assumption that a single, relatively short preparatory teaching unit for physicians who are at the beginning of their clinical training could lead to a significant difference in performance between the two study groups is therefore very optimistic...

Debriefing is an essential element of simulation training and regarded as an important prerequiste for learning success. However, in the study design, debriefing is only planned if the patient has been declared dead prematurely. What is the reason to exclude all other participants from debriefing?

Further remarks:

The Files S2 and S3 are not provided within the manuscript.

Thank you very much for the opportunity to examine this very exciting topic for medical education and training!

Reviewer #2: Line 105: Phase 2 to 3

Line 110-111: information on the list participants preparation, allocation concealment blinding is to be stated.

Line 112-113: the type of randomization is to be stated.

The detail of the intervention and control is be provided e.g. content, topics, time to complete the entire course (e-earning, VR) over the period of two-week

The assessment period/time point is to be mentioned.

Line 141-142: Secondary objectives are too many. Suggest to reduce and combine some of the objectives

Line 160: Sample size calculation not clear. More information such as primary outcome, 1 tailed or r tailed test, measures estimate, formula or software was used are to be stated.

Line 208-209: the sentence requires revision.

Line 292: effect size index, 95%CI are be added where applicable.

The statistical software including publisher name and its version used in the analysis is to be stated.

Flow chart of the study is to be provided.

Line 209: ERC, ALS are to be spelled out.

Line 242: the number of raters and who are they to be clearly stated.

Line 258: the sentence requires revision.

Page 13: the person who performed the eye tracking is to be stated.

Line 298: the reason/rationale to use non parametric tests is to be mentioned.

Line 307 & 309: the sentence requires revision.

Line 309: one-tailed or two-tailed test for Fisher's exact test is to be stated.

Line 323: what about missing data for secondary outcome? if the missing data is significant, what is the solution to address it? The variables with the possible censored data (if any) other than primary outcome is to be highlighted.

Line 324: the sentence ‘boxplots and scatterplots could be used to illustrate calculation’ is misleading.

Language version of the tools/inventories used in the study are to be stated.

The write-up on the statistical methods could be enhanced by organizing it in a more systematic manner and flow.

7. PLOS authors have the option to publish the peer review history of their article (what does this mean?). If published, this will include your full peer review and any attached files.

Reviewer #1: No

Reviewer #2: No

---

## [Author Response · Author response to Decision Letter 0]

20 Nov 2024

Dear reviewers,

we would like to thank you for your constructive feedback and send you our responses to your comments on the submitted manuscript. 

Please note that page and line numbers refer to the document containing the markups.

Reviewer #1

Simulation training is regarded as a major element of patient safety and preparation for time-critical life-threatening medical emergencies, i.e. severe trauma and cardiac arrest. Despite the high benefit of in-situ simulation training in such scenarios, it is rarely carried out due to the complexity of the scenarios to be practiced and the high equipment and personnel expenditure required. Extended reality (XR) is becoming increasingly important in medical simulation training as it allows immersive realistic training scenarios undertaken with less expensive equipment. Virtual reality (VR) is currently the most common variant of XR in this context.

The presented study protocol intends to compare the effectiveness of VR-based and E-learning-based preparation for in-situ training of in-hospital traumatic cardiac arrest (TCA)- a rare, but extremely challenging, scenario. The hypothesis is that this blended-learning approach is of superior educational value and could improve the management skills in TCA.

Strengths of the paper:

The criteria for recruitment and eligibility, the randomization procedure, the assignment of the participants to the comparison groups, the primary and secondary endpoints of the study and the chosen measurement methods are well chosen und clearly described. The secondary endpoints “influence of gender and personality” could give valuable insight into potential additional requirements or needs of important user subgroups. Tables and figures are clear and easy to interpret. The SPIRIT checklist was used to report all study protocol items. The protocol fulfills the respective legal, ethical and security requirements, was approved by the local ethics committee and the study has been registered in clinicaltrial.gov. In addition, the protocol will also be reviewed by the data protection commission. The statistic procedures are well described, suitable and possible confounders are adequately taken into account. The cited literature sources are appropriate and mostly up to date.

Points where I would like to suggest a revision:

The authors state that (Page 3, lines 81 ff) the preparation training will “equip learners with the skills and knowledge necessary for in-person simulation training and to improve their performance in TCA management skills”. Unfortunately, only scarce information is provided about the specific learning objectives and contents, as well as the media technology and didactic implementation of the e-learning and VR training. The presentation of these points is particularly important because the treatment of traumatic cardiac arrest is a very complex task that requires a) theoretical knowledge of the mechanisms and treatment principles of trauma and resuscitation, b) manual or technical skills to eliminate the underlying causes, e.g. tension pneumothorax, and c) excellent management or non-technical skills. This makes it difficult to answer the questions whether learning objectives and content of the two methods are comparable. I therefore would recommend to describe the intended interventions in more detail. More details would also enhance the understanding of the setting of the in-situ simulation training (details see below).

Related to the chosen endpoints and their significance I would suggest a more detailed discussion of potential limitations (details see below).

Thank you for your valuable and constructive feedback. Accordingly, we have now defined and formulated the points mentioned in more detail (see the individual sub-points for details). At this point we would like to mention that both the e-learning and the VR software are still being developed and finalized in an inter-professional exchange. As a result, we can only comment on the defined learning content and objectives, but not yet on the final technical implementation.

Page 5, lines 115 ff:

“Participants will then complete either the e-learning course (control group) or the VR training

(intervention group) over a two-week period, both focusing on the same content”.

E-Learning:

Which kind of media are used? As trauma shockroom management depends not only on clinical skills but also on communication and management skills: To what extent are these “non-technical skills” addressed? Does the e-learning contain intervening questions that encourage participants to self-reflect? Are there other interactive elements, i.e. clicking on answer options, e.g. to select a cause of cardiac arrest or to decide on a specific diagnostic or therapeutic measure, or abstracted manual actions such as applying a pelvic sling, insertion of a chest tube?

VR training:

Will a head-mounted display (HMD) be used? If so, what type? Can you present the technical specs (i.e. as supplement data)? Does the VR “training” represent training in the narrower sense of the word, i.e. is it interactive and allows active clinical decisions, or is does it consist of mere passive tracking of the content presented? If the former is true, could you describe the types of interaction?

Details are now explained in more detail. Additionally, we would like to note that while elements of non-technical skills are incorporated into both training approaches, the primary focus remains on procedures and adherence to guidelines. As a result, only these components will be evaluated in the final assessment.

Changed parts of the manuscript:

Page 4, line 110ff: Participants will then work independently through the guidelines provided.

Page 8, line 219ff: Participants will engage in case-based learning using a variety of case vignettes offered in both e-learning and VR environments. The cases used will focus on the same medical content and are designed to be practiced in a structured way. The overarching goal of both learning modalities is to “apply“ the necessary life-saving action sequences within the framework of a TCA, according to Bloom's taxonomy [33,34].

The key features of each learning modality are outlined below:

E-learning: This course includes step-by-step descriptions of action sequences, along with videos demonstrating specific actions and communication examples. Interactive components – such as mapping exercises, quizzes, and comprehension questions – help reinforce learning. Participants can explore images with expandable content, follow action sequences in chronological order, and respond to questions that encourage both clinical decision-making and self-reflection.

VR: In this interactive virtual environment, participants play through scenarios, assign specific tasks to virtual team members, and make clinical decisions. Virtual team members provide communication examples with feedback. At the end of each scenario, participants receive feedback on their adherence to guidelines, along with questions for self-reflection. 

Page 8, line 240ff: A head-mounted display (HTC Vive Focus 3) will be used for the single-player VR scenarios, with an additional trainer operating from a computer that serves as the control unit to manage the scenarios.

Page 6, lines 122 ff:

How long is the interval between completion of Phase 2 and Start of Phase 3?

The corresponding interval is now explained in greater detail. 

Changed parts of the manuscript:

Page 5, line 134: Within one week of completing the VR- or e-learning preparation courses, participants will attend… 

Page 7, Table 1:

Several secondary endpoints are measured to assess the participant´s subjective impressions and attitudes (i.e. level of enjoyment, learning process, self-confidence). Are these measurements based on validated test instruments? If not, what reasons spoke against? There are widely recognized assessment tools in this area, i.e. TEI (Training Evaluation Inventory).

In response to this and also reviewer #2's comment, the corresponding secondary objectives were shortened or summarized. Furthermore, the validated TEI test instrument was incorporated into the study protocol in lieu of individual questions.

Changed parts of the manuscript:

Page 5, line 137ff: In addition, each participant will be required to complete another questionnaire immediately following the assessment session (including the German version of the training evaluation inventory (TEI)

Page 6, Table 1: 

Primary To evaluate the difference in time to critical action #2 between randomized groups Time (seconds) Expert assessment of video recordings of the assessment session: time from the onset of cardiac arrest to critical action #2 (as defined in Table 2)

Secondary To evaluate the difference in cognitive load between randomized groups 0 to 100 points Assessed immediately following the assessment session

Global NASA TLX score

NASA TLX score per objective (mental demand, physical demand, temporal demand, performance, effort, frustration)

Secondary To evaluate the difference in learning outcomes and training design using the TEI between randomized groups 5-point Likert scales Training outcomes (subjective fun, perceived usefulness, perceived difficulty, subjective knowledge growth, attitude towards training) and training design (problem-based learning, activation (of prior knowledge), demonstration, application, integration) will be measured using a 45-items inventory (I do not agree; I rather not agree; neither nor; I rather agree; I agree very much)

In addition, secondary objectives assessing individual questions on subjective impression have been removed and replaced with the TEI;

Correlations on video game experience have been removed and will only be assessed as a baseline characteristic using a 5-point Likert scale - see page 14, line 396ff: …, as well as participants’ self-reported VR- and non-VR video game skill sets (rated on 5-point Likert scales);

The system usability scale has been removed;

Secondary objective: “Critical action #1: time from the onset of cardiac arrest to the start of chest

compressions (as defined in Table 2)”.

According to current resuscitation guidelines, chest compression is of secondary importance to other measures in traumatic cardiac arrest: German S3 updated guideline on the treatment of severe/multiple injuries: “Recommendation 2.3.4 [translated into English]: “Resuscitation of traumatic cardiac arrest should focus on the immediate, simultaneous treatment of reversible causes and takes priority over chest compressions”. ERC guidelines 2021 (Lott et al.): „In cardiac arrest caused by hypovolaemia, cardiac tamponade or tension pneumothorax, chest compressions are unlikely to be as effective as in normovolaemic cardiac arrest and may reduce residual spontaneous cardiac output. Therefore, chest compressions take a lower priority than addressing the reversible causes. Chest compressions must not delay immediate treatment of reversible causes.”

Could this potentially limit the significance of the primary endpoint “start of chest compressions”?

We removed critical action #1 because it is of lower priority in the context of traumatic cardiac arrest and therefore not the focus of the performance assessment.

Changed parts of the manuscript:

Table 1, Table 2

Page 7, Table 1: Secondary objectives: “To evaluate the difference in gaze behavior using eye-tracking technology between randomized groups”.

How is the quality of eye tracking (i.e. spatial precision and spacial accuracy) ensured?

Details are now explained in more detail. 

Changed parts of the manuscript:

Page 13, line 380ff: Moreover, prior to the commencement of each assessment scenario, the eye-tracking glasses will be calibrated on the respective participant. Additionally, attention will be paid to ensure that the lighting and environmental conditions are as identical as possible, thus ensuring optimal eye-tracking quality in terms of spatial precision and spatial accuracy.

Page 8, line 151: Inclusion criteria: “Only people who do not need eyeglasses for using VR”

How can this be interpreted and how do you practically handle this problem? Do interns wearing eye glasses and willing to participate in the study first test a VR system / HMD before they definitively decide to participate or to withdraw from the trial? Since the high percentage of people dependent on visual aids, this is not a mere theoretical question….

As visual impairments can be compensated for in the planned hardware solution, this exclusion criterion is no longer relevant and has been removed.

Changed parts of the manuscript:

Page 7: Exclusion criteria

Page 9, lines 170-171: Additional VR orientation module: What is the duration of this module? Will it be mandatory for all participants In the VR group?

Details are now explained in more detail. 

Changed parts of the manuscript:

Page 8, line 236ff: To ensure optimal efficiency during the VR training, an additional mandatory orientation module of 20 to 30 minutes, to be played once before using the VR scenarios, will be provided to all participants in the VR group.

Page 9, lines 184-186:

Is the duration of the familiarization phase standardized? Do the participants familiarize themselves individually or as a whole group? Is there an instructor who assists during this phase i.e. explaining the use of the manikin functions?

Details are now explained in more detail. 

Changed parts of the manuscript:

Page 9, line 253ff: Prior to commencing the manikin-based simulation scenario, participants will be given 15 to 20 minutes to familiarize themselves with the room, the environment, the manikin's functions, equipment, and provided medications. In the assessment scenario, all participants will use the same male adult manikin, which will display clinical signs such as cyanosis, breath sounds, and pulses to aid in clinical decision-making. An instructor will be present to assist the group during this phase.

Page 9, lines 187-188 and page 10, lines 200-205:

Does the virtual patient in the scenario already present with a cardiac arrest (i.e. at the moment of the virtual hand-over by the EMS team) or does the cardiac arrest develop after a predetermined short latency?

Medical scenarios: Is there always a single injury/medical problem leading to the cardiac arrest or maybe also a combination of underlying causes?

Details are now explained in more detail. 

Changed parts of the manuscript:

Page 9, line 258ff: Both groups will face the same standardized assessment scenario: a trauma patient with a single predefined underlying condition that progresses to an IHCA after a predefined short latency period. Each scenario will have a duration of 15 minutes.

Does the “high-fidelity” manikin used in the study display clinical signs like cyanosis, recap time, breath sounds, subcutaneous emphysema, obstructed airway (which could facilitate clinical decision making)?

Is a single type of manikin used or do the manikins vary, i.e. according to scenario? What age group (Child? adolescent? adult? Young adult? old adult?) and sex does the manikin used in the study represent?

Details are now explained in more detail. 

Changed parts of the manuscript:

Page 9, line 255ff: In the assessment scenario, all participants will use the same male adult manikin, which will display clinical signs such as cyanosis, breath sounds, and pulses to aid in clinical decision-making. An instructor will be present to assist the group during this phase.

Page 10, lines 215-217:

“These critical actions are interconnected, and the primary outcome is, therefore, the time taken to execute the most critical action in the assessment scenario (critical action #3).”

In practice, when carrying out life-saving measures, it will take some time until a) they have been carried out after being initiated and b) until the success of the measures can be assessed. Only then can further decisions be made and further measures ordered. Consecutively

a) the term “time taken to execute” may be difficult to interprete or be misleading and

b) the problem of eventually simultaneously necessary actions is not addressed nor solved

The corresponding paragraph

---

## [Decision Letter · Decision Letter 1]

17 Dec 2024

Enhancing trauma cardiopulmonary resuscitation simulation training with the use of virtual reality (Trauma SimVR): Protocol for a randomized controlled trial

PONE-D-24-24609R1

Dear Ms. Hafner

We’re pleased to inform you that your manuscript has been judged scientifically suitable for publication and will be formally accepted for publication once it meets all outstanding technical requirements.

Kind regards,

Anne Lee Solevåg, M.D., Ph.D.

Academic Editor

PLOS ONE

Additional Editor Comments (optional):

Reviewers' comments:

Reviewer's Responses to Questions

**Comments to the Author**

1. Does the manuscript provide a valid rationale for the proposed study, with clearly identified and justified research questions?

Reviewer #1: Yes

Reviewer #2: Yes

2. Is the protocol technically sound and planned in a manner that will lead to a meaningful outcome and allow testing the stated hypotheses?

Reviewer #1: Yes

Reviewer #2: Yes

3. Is the methodology feasible and described in sufficient detail to allow the work to be replicable?

Reviewer #1: Yes

Reviewer #2: Yes

4. Have the authors described where all data underlying the findings will be made available when the study is complete?

Reviewer #1: Yes

Reviewer #2: Yes

5. Is the manuscript presented in an intelligible fashion and written in standard English?

Reviewer #1: Yes

Reviewer #2: Yes

6. Review Comments to the Author

You may also provide optional suggestions and comments to authors that they might find helpful in planning their study.

Reviewer #1: Dear authors,

Thank you for revising the manuscript!

The points noted have all been carefully revised.

I wish you every success in carrying out the study!

Reviewer #2: The authors have diligently and thoroughly addressed all comments. There are no further comments.

7. PLOS authors have the option to publish the peer review history of their article (what does this mean?). If published, this will include your full peer review and any attached files.

Reviewer #1: No

Reviewer #2: No

---

## [Editor Report · Acceptance letter]

30 Dec 2024

PONE-D-24-24609R1 

PLOS ONE

Dear Dr. Hafner, 

I'm pleased to inform you that your manuscript has been deemed suitable for publication in PLOS ONE. Congratulations! Your manuscript is now being handed over to our production team.

Kind regards, 

on behalf of

Dr. Anne Lee Solevåg 

Academic Editor

PLOS ONE